# OpenReview forum: "Get more for less: Principled Data Selection for Warming Up Fine-Tuning in LLMs"
_ICLR.cc/2024/Conference — ICLR 2024 poster_

### Official Review · Reviewer_nPTw · 2023-10-14

**Soundness:** 4 excellent
**Presentation:** 3 good
**Contribution:** 3 good
**Rating:** 8
**Confidence:** 4

**Summary:**

This work presents "pre-fine-tuning" as a method to utilize open, unlabeled data to enhance task adaptation of LLMs. It proposes a new strategy, GOT-D, which adjusts the pre-training distribution closer to the target distribution via optimal transport. Evaluations on various language tasks show the advantages of GOT-D in performance as well as efficiently handles millions of samples.

**Strengths:**

1. The idea of pre-fine-tune is novel and interesting. The proposed GOT-D is well motivated, theoretically principled, and scalable.
2. Extensive experiments show the superiority of the proposed method
3. Overall, the paper is well-organized and comprehensible, despite the inclusion of some mathematical derivations.

**Weaknesses:**

I don't see any major weakness of this work, but I'm not fully convinced by the idea of pre-fine-tune. For example, what if we jointly fine-tune the model on both the selected unlabeled dataset and the target dataset, like this prior work does [1]?

In addition, it would be better to include in-depth study on the scaling of pre-fine-tune. For example, if we keep increasing the data for pre-fine-tune, would it lead to a reduction of need for target data?

[1] Improved Fine-Tuning by Better Leveraging Pre-Training Data

**Questions:**

See weakness

---

> ### Author Response · Authors · 2023-11-21
> **Weaknesses-1: What if we jointly fine-tune the model on both the selected unlabeled dataset and the target dataset**
>
> **Weaknesses-1: What if we jointly fine-tune the model on both the selected unlabeled dataset and the target dataset, like this prior work does [1]?**
>
> **Re.:** The authors are grateful to the reviewer for the dedicated work and positive comments. This is an interesting question and this prior work actually provided inspiration during the early development of this work. There are several reasons we are doing differently from the setup of this work.
>
> For NLU tasks in Section 3.3/3.4, the fine-tuning data is labeled and our pre-fine-tuning selects unlabeled data from raw corpora. Thus, the fine-tuning is naturally conducted in a sequential manner. It is not straightforward to combine unlabeled/labeled data in a joint fine-tuning scheme. This semi-supervised learning problem on its own could be an interesting direction for research exploration.
>
> The NLG task in Section 3.2 as well as the additional experiments are  "zero-shot" as its fine-tuning is only conducted in an unsupervised/self-supervised manner. The pre-fine-tuned model can be directly deployed to downstream tasks without further fine-tuning. We consider the target data used for data selection as the **“validation data”**. The validation data is usually considered identically distributed with the test data and cannot be used for training the model. This setting is typical in data selection/valuation works.

---

> ### Author Response · Authors · 2023-11-21
> **Weaknesses-2: it would be better to include in-depth study on the scaling of pre-fine-tune**
>
> **Weaknesses-2: it would be better to include in-depth study on the scaling of pre-fine-tune. For example, if we keep increasing the data for pre-fine-tune, would it lead to a reduction of need for target data?**
>
> **Re.:** The authors agree that the experiment mentioned by the reviewer would be desirable. Figure 1 provides this exact result. We show that on CoLA task from GLUE benchmark, conducting pre-fine-tuning with selected unlabeled data could help reach the same downstream performance with much fewer labeled data.
>
> In light of the direction pointed out by the reviewer, we conducted ablation studies on pre-fine-tuning data sizes with our available resources. The results are presented in Table 16 at the end of Appendix. **Performance improvements appear to comply with a log-linear scaling law relationship with pre-fine-tuning data size with the performance recording constant improvement every time the fine-tuning data budget is doubled.**

---

> ### Author Response · Authors · 2023-11-22
> **Rebuttal period ending–we anticipate your feedback!**
>
> Dear Reviewer nPTw,
>
> As the rebuttal/author discussion period is closing, we sincerely look forward to your feedback. The authors are deeply appreciative of your valuable time and efforts spent reviewing this paper and helping us improve it.
>
> It would be very much appreciated if you could once again help review our responses and additional results and let us know if these address or partially address your concerns and if our explanations are heading in the right direction.
>
> Please also let us know if there are any further questions or comments about this paper. We strive to consistently improve the paper and it would be our pleasure to have your precious feedback!
>
> Kind Regards,\
> Authors of Submission530

---

### Official Review · Reviewer_cFNr · 2023-10-31

**Soundness:** 2 fair
**Presentation:** 2 fair
**Contribution:** 3 good
**Rating:** 5
**Confidence:** 3

**Summary:**

This paper tackles data selection for *pre-finetuning* LLMs in order to improve performance and reduce the need for manually curated fine-tuning data. The authors differentiate this setting from continued-pretraining, which has been explored in past works, because the data selection budget is assumed to be smaller. Their proposed method is then motivated by the key observation that the common practice of selecting data based upon distribution matching may lead to many redundant selections of examples already well-learned from pre-training. Instead, they choose the examples which most effectively nudge the training distribution closer to the target via optimal transport. They then test their method on toxicity prevention and language understanding tasks and compare to previous baselines.

**Strengths:**

Overall, the authors tackle a well-motivated problem and the high-level idea to avoid redundancy when selecting for fine-tuning data seems quite interesting/novel. Their method runs efficiently (compared to previous works), due to only requiring solving one OT problem, and holds promise for scaling up to larger datasets. The empirical results on the toxicity reduction are also impressive.

**Weaknesses:**

The two main areas of improvement for this paper are (1) presentation, particularly reducing reliance on the Appendix for key details; (2) more thorough experiments/ablations to assess the core aspect of their method (i.e., to me, this is the idea that it is important to "select data that is not redundant with pre-training")

(1) Presentation
- There were several details that I think are important to share more up front. For instance, in the experiments, it should be clear both (a) what the candidate datasets are; (b) what feature space/function is being used to calculate costs in OT. I also think the different pre-processing steps different methods do to the candidate set should be mentioned.
- While having an extended related work in the Appendix is fine, it would nice if more of this content could be covered in the main.
- Some aspects of the notation could also be cleaned up (see Questions below)
- The All Domains baseline is mentioned on pg. 6 before being defined.

(2) Experiments
- For the experiments in Sec 3.2-3.3, the improvements from GOT-D over previous baselines don't seem to be that much in most cases (i.e., within standard deviations)? Perhaps the authors can comment on significance here?
- Currently, it seems there are several differences between DSIR and GOT-D besides just that one does distribution matching and the other finds the most informative examples (given pre-training distribution). Particularly, they both involve preprocessing the candidate pool in and use different feature representations to perform selection (i.e., DistillBERT v.s. n-gram). I think more controlled ablations of these other factors are important to ablate what precisely causes GOT-D to perform better.
- In the BERT experiments, do DSIR and DAPT/TAPT use the same pre-processing procedure to select relevant domains as described for GOT-D? I'd be curious what happens if this step is left out for GOT-D.
- It would be nice to show some qualitative examples of what kinds of examples get selected by GOT-D but are not as emphasized in distribution matching approaches (e.g. DSIR).

**Questions:**

(1) In the introduction it is mentioned that *"Intuitively, fine-tuning a pre-trained model with such samples would boost its performance on the target dataset. We prove the validity of this intuition under certain assumptions, thereby setting our method on a solid theoretical foundation."* To me, this suggested there would be a formal theorem statement in the text that explicitly outlines assumptions and a guarantee? Section 2 does describe and link together some existing results in prose but it's not as clear what has been newly proven?

(2) In the toxicity reduction experiments, am I correct in understanding that no methods did the second round of task-specific fine-tuning? If so, I don't think this would change the core conclusion but I think it is still important to include a baseline which trains on just the initial 2.5K target examples. Similarly, it would be nice to confirm that the improvements of GOT-D over other methods persists after doing the fine-tuning round on these examples.

(3) As a general comment, the distinction between the terms "pre fine-tuning" and "continued pre-training" remains a bit unclear to me. My understanding from the paper is the same methods could be applied to both but the main difference is just how much data is involved. But in some places "pre-training" is still used a bit interchangeably (e.g.. *"This experiment involves two stages: pre-training over selected data and then fine-tuning over the downstream task."* in Sec 3.2, pg. 7). What is the motivation for defining the new term as opposed to saying something akin to "continued pre-training with limited budget"?

(4) Miscellaneous
- It seems sometimes $D_S, D_R, etc.$ are used interchangeably to represent (finite) datasets and distributions. Perhaps the authors can make this distinction clearer?
- In Eq. (3), should the last term of the expansion include $\lambda \cdot (D_U - D_S)$ instead of just $\lambda \cdot (D_U)$?
- The version of TAPT here seems to be the "curated-TAPT" described in Gururangan et al. rather than "TAPT", which I've more typically seen as just training on the unlabeled end-task inputs. Perhaps the authors can add a clarifying note here, or better, include results for both versions of TAPT.
- In Table 3, it seems TAPT should be bolded for RCT. Also the text right above mentions "All domains" but it seems to be missing from the table?

---

> ### Author Response · Authors · 2023-11-21
> **A general response to Reviewer cFNr**
>
> The authors are of great appreciation for the profoundly helpful feedback and comments from Reviewer cFNr. The review is exceptionally constructive, detailed, insightful, and kind. It is fortunate for this work to receive this careful review and the authors are deeply encouraged by this experience.
>
> The authors agree with the revision direction pointed out by the review and are extremely motivated to respond to and fix all the issues.
>
> At this moment, we have made major revisions to the manuscript.
>
> - We have revamped the main methodology section (Section 2.3), replacing the previous derivation-with-elaboration with lemma/theorem with proof and remarks to improve its clarity and general rigor.
>
> - We made a number of local edits to improve the presentation, adding details, and fixing errors to improve the general clarity and readability.
>
> - We added a whole set of additional experiments with controlled settings to ablate the effect of multiple design choices. We also added visualizations to facilitate insights into the data selected by each method.
>
> We have more edits to make. Specifically, we are working on simplifying the notation system in Section 2 and improving its intuition, adding the new experiments to the main paper, adding more discussion on related works and experiment details to the main paper and reducing the reliance on the Appendix, adding sufficient explanations of the significance and takeaway for the experiments in sections 3.3/3.4, etc. Some of the planned edits require making more changes to the current content to make space for all the needed revisions.
>
> The authors are committed to improving the overall quality of this paper and are working dedicatedly for it.

---

> ### Author Response · Authors · 2023-11-21
> **Weaknesses-1, Presentation**
>
> **Weaknesses-1, Presentation:**
>
> **(i) There were several details that I think are important to share more up front.**
>
> > "For instance, in the experiments, it should be clear both (a) what the candidate datasets are; (b) what feature space/function is being used to calculate costs in OT. I also think the different pre-processing steps different methods do to the candidate set should be mentioned. While having an extended related work in the Appendix is fine, it would nice if more of this content could be covered in the main."
>
> **(ii) Some aspects of the notation could also be cleaned up (see Questions below)**
>
> **(iii) The All Domains baseline is mentioned on pg. 6 before being defined.**
>
> **Re.:** **(i).** Thanks for the insightful feedback. The authors agree with the direction pointed out by the reviewer and are committed to making revisions accordingly.
>
> Currently, due to the many contents and volume of experiments, we provide a comprehensive presentation for all the relevant details–models, datasets, implementation, processing steps, etc. in Appendix C. We agree that it would provide a better presentation to include some key details right in the main paper. We are working toward that direction. Currently, we have added more details to the main text where the space allows. As mentioned in the previous response, we are currently working on revamping the experiment sections, condensing the current contents and making space for the revisions. It is our priority to include more key information in the main paper as suggested by the reviewer. We agree it should improve the readability of the paper and its overall quality.
>
> **(ii).** We have revamped the main methodology section and remodeled it in the style of lemma/theorem with proof and remark in an effort to improve its clarity and general rigor.
>
> We are working on simplifying the notation system used in this work and making it more intuitive and straightforward. We will make sure the mentioned notation issues are taken care of during this process. We wish this could further improve the readability of the manuscript, avoid potential confusion, and make it more accessible to broader readers.
>
> **(iii).** Thanks for the catch. Apologies. We have fixed this and also checked other contents for inconsistencies.

---

> ### Author Response · Authors · 2023-11-21
> **Weaknesses-2, Experiments: (i). Comment on the significance of performance improvements in Section 3.3/3.4**
>
> **Weaknesses-2, Experiments: (i). Comment on the significance of performance improvements in Section 3.3/3.4**
>
> **Re.:** The authors did realize this presentation issue not long after submitting the manuscript.
>
> 8 NLU tasks with a pre-defined domain (DAPT tasks, Section 3.3/Table 2, 3) are first presented in DAPT paper, where they show continued pre-training RoBERTa model for 2\~8B tokens on a target domain (equivalent to **20M\~80M** samples with 100-token length) could improve the performance on the tasks in this domain by **2.3%** on average (the improvement is less than 2% on 5 tasks). DSIR achieves an average of **1.56% improvements after continued pre-training on 25M samples.**
>
> In this work, we show that favorable performance improvements can be established with a **much smaller** fine-tuning budget if the data is selected appropriately. We achieved an average of 1.13% improvement after fine-tuning with only 150k samples **(~0.5% of the data budget for continued pre-training) in Table 2 and a 0.76% improvement with 50k samples in low-resource scenarios in Table 3. It can be seen from the tables that the rate of performance improvements for GOT-D significantly **outpaces** that of baseline methods, highlighting the efficacy and data efficiency of the proposed method.
>
> The results for 8 NLU tasks without a pre-defined domain (GLUE tasks Section 3.4/Table 4) are meant to be interpreted in a slightly different way. GLUE tasks are designed to measure the **general capability** of a language model. These tasks are specifically selected to measure very different aspects of the language model and it is hard to achieve a high score in all aspects at the same time.
>
> We conducted this experiment as a test to see whether light fine-tuning can improve the scores on this kind of benchmark. DSIR achieves a **2%** improvement in GLUE score by curating pre-training data and training a BERT model from scratch for **51.2M** samples. **It is expected that performance improvements can be difficult.** As mentioned in the discussion in the paper, the proposed light pre-fine-tuning is more beneficial for tasks that benefit from certain task-relevant knowledge, rather than the ones relying on the fundamental capability of the model. It is an interesting result for us to see that light pre-fine-tuning with the proposed method achieves more than 1% performance improvement on the GLUE benchmark.
>
> It is our responsibility not being able to provide sufficient discussion after the experiments to elaborate on the **takeaway** for each result.
>
> **Besides, we have added a new set of experiments for pre-fine-tuning GPT models for zero-shot classification tasks, where the performance improvements of our method are 6.6%~13.9% after pre-fine-tuning with 40k samples.**

---

> ### Author Response · Authors · 2023-11-21
> **Weaknesses-2, Experiments: (ii). ...several differences between DSIR and GOT-D**
>
> **Weaknesses-2, Experiments: (ii). ...several differences between DSIR and GOT-D... both involve preprocessing the candidate pool in and use different feature representations to perform selection (i.e., DistillBERT v.s. n-gram). ...more controlled ablations of these other factors are important to ablate what precisely causes GOT-D to perform better.**
>
> **Re.:** We agree with the reviewer that there exists several differences in processing steps and it is of interest to ablate on their effects to better understand this problem.
>
> This has also been one of our concerns during the development of this work, and after careful discussions, we had decided not to change the processing pipelines since this is a key part of the main philosophy of each method. The method would have a different name if we alter its processing steps/embedding scheme, etc. For example, one of the main efforts of the DSIR paper is to show that the distributional distance on n-gram space is a valid proxy for downstream performance on target tasks so that they can use this simple feature to enable large-scale selection for pre-training data. Yet, it is also true that it is desirable to ablate the effect of these design choices when comparing it with other methods.
>
> The implementation of DSIR is highly integrated and we are not able to replace its embedding scheme in a straightforward way. We conducted more controlled studies in the additional experiments. In the zero-shot classification task with GPT-2 XL, we ablate on the effect of choices of embedding space for computing OT distance. We tested embedding samples with distilled-BERT, sentence-transformer, and BERT-tokens. Lightweight and fast, the popular sentence-transformer uses a pre-trained all-MiniLM-L6-v2 model with 22M parameters as the backbone. It embeds up to 6 million samples/hours on a single GPU and is sometimes considered a 'default' option for sentence embedding in many NLP tasks. Token space isn't a proper embedding for OT (e.g., the distance on token space is not invariant to paraphrase). We are only listing it here for comparison. As can be seen from Table 17, the performance of sentence-transformer is mostly on par with distilledBERT. It suggests **the choice of embedding space might not be a critical part of our proposed data selection pipeline and any reasonable embedding space should work.**
>
> **Besides, we also provide visualization of samples selected by each method to facilitate insights into the experiment results.** In the additional experiments with the “AG News” task where all the settings are best aligned, we conduct visualization of samples selected by each method and the distribution of target task data and pretraining/candidate data, offering insights on how the proposed method works differently from the distribution-matching approaches. As shown in Figure 16, the comparison shows a clear contrast. **Both DSIR and TAPT/c select samples that match the distribution of the target task data which has a high overlapping with the distribution of the candidate data where the model is already pre-trained on.** Thus, with such a small data budget, the information gain provided from pre-fine-tuning on these samples is naturally marginal. In contrast, **GOT-D selects predominately formal business news which is highly underrepresented in the candidate dataset but important for the target task.** Pre-fine-tuning the model with these samples provides a more direct benefit in aligning the model with the target tasks, **effectively validating the idea of this work.**

---

> ### Author Response · Authors · 2023-11-21
> **Weaknesses-2, Experiments: (iii). pre-processing procedure in BERT experiments**
>
> **Weaknesses-2, Experiments: (iii). In the BERT experiments, do DSIR and DAPT/TAPT use the same pre-processing procedure to select relevant domains as described for GOT-D? ...what happens if this step is left out for GOT-D.**
>
> **Re.:** The authors acknowledge the responsibility for the processing scheme not presented straightforwardly enough.
>
> DSIR is designed for selecting data from all domains, which is one of its advertising advantages and also why it chooses the simpler n-gram feature space. DAPT/TAPT inherently requires a curated domain dataset, so its selection is always from the relevant domain dataset. Similar to the previous response, we are reluctant to change its processing steps. In the DSIR paper, the authors also conduct comparisons with DAPT under the same setting.
>
> These BERT experiments are resource-intensive. A sizeable number of models need to be fine-tuned considering the number of tasks, baselines, data sizes, repetitions, and many ablations for design choices. Instead, we ablate the effect of this processing step in the additional experiments.
>
> In the zero-shot classification task with GPT-2 XL, since the model is pre-trained on a single domain–OpenWebTextCorpus(OWTC),  **we use the same data as the candidate dataset and all baseline methods selected from the same pool of candidate data without additional processing.** The setting is well-aligned and consistent with assumptions in the development of our method. The proposed method achieves a performance gain of 13.9% after pre-fine-tuning with 40k samples, visibly outperforming DSIR and TAPT.
>
> This is also the case for the NLG task with GPT-2 base in Section 3.2.

---

> ### Author Response · Authors · 2023-11-21
> **Weaknesses-2, Experiments: (iv). qualitative examples of what kinds of examples get selected**
>
> **Weaknesses-2, Experiments: (iv). It would be nice to show some qualitative examples of what kinds of examples get selected by GOT-D but are not as emphasized in distribution matching approaches (e.g. DSIR).**
>
> **Re.:** The authors agree that it will be desirable to direct compare the samples selected by each method and random selection from pre-training data. We added this to Appendix E.1.
>
> In the additional experiments with the “AG News” task where all the settings are best aligned, we conduct visualization of samples selected by each method and the distribution of target task data and pretraining/candidate data, offering insights on how the proposed method works differently from the distribution-matching approaches. **As shown in Figure 16, the comparison shows a clear contrast. Both DSIR and TAPT/c select samples that match the distribution of the target task data which has a high overlapping with the distribution of the candidate data where the model is already pre-trained on.** Thus, with such a small data budget, the information gain provided from pre-fine-tuning on these samples is naturally marginal. In contrast, **GOT-D selects predominately formal business news which is highly underrepresented in the candidate dataset but important for the target task.** Pre-fine-tuning the model with these samples provides a more direct benefit in aligning the model with the target tasks, effectively validating the idea of this work.

---

> ### Author Response · Authors · 2023-11-21
> **Question-1: formal theorem statement**
>
> **Question-1: ...this suggested there would be a formal theorem statement in the text that explicitly outlines assumptions and a guarantee?**
>
> **Re.:** The authors agree that it is important and needed to present the proposed method and its development in a more formal and rigorous manner.
>
> **We have revamped the main methodology section (Section 2.3), replacing the previous derivation-with-elaboration with lemma/theorem with proof and remarks to improve its clarity and general rigor.**

---

> ### Author Response · Authors · 2023-11-21
> **Question-2: In the toxicity reduction experiments... no methods did the second round of task-specific fine-tuning**
>
> **Question-2: In the toxicity reduction experiments... no methods did the second round of task-specific fine-tuning... it is still important to include a baseline which trains on just the initial 2.5K target examples... to confirm that the improvements of GOT-D over other methods persists after doing the fine-tuning round on these examples**
>
> **Re.:** Thanks for the suggestion. The NLG task in Section 3.2 as well as the additional experiments are  "zero-shot" as its fine-tuning is only conducted in an unsupervised/self-supervised manner. **The pre-fine-tuned model can be directly deployed to downstream tasks without further fine-tuning.** We consider the target data used for data selection as the “validation data”. **The validation data is usually considered identically distributed with the test data and cannot be used for training the model.** This setting is typical in data selection/valuation works.
>
> As a comparison, these validation data are prepared in the same way as RTP–using Perspective API to evaluate the toxicity of OWTC samples and select a clean/toxic batch. Fine-tuning on the 2.5k clean batch might yield a similar result to RTP with 2.5k data, which will not be as good as pre-fine-tuning the model on the data selected by the proposed methods.

---

> ### Author Response · Authors · 2023-11-21
> **Question-2:  distinction between the terms "pre fine-tuning" and "continued pre-training"**
>
> **Question-2: As a general comment, the distinction between the terms "pre fine-tuning" and "continued pre-training" remains a bit unclear...What is the motivation for defining the new term as opposed to saying something akin to "continued pre-training with limited budget"?**
>
> **Re.:** Thanks for the suggestion. This is actually a bit of a tricky issue that has been on the minds of the authors throughout. The simple reason is all the alternatives we can think of may potentially cause confusion in other contexts.
>
> Yes, the proposed scheme is essentially continued pre-training with a (much) smaller budget. The tricky part is that continued pre-training is already very well-known to be associated with a scope that is orders of magnitudes more than ours.
>
> Continued pre-training provides techniques for **pre-trained model builders** to build separate pre-trained models for separate tasks given ample compute and data budgets. It is perceived to be part of the pre-trained model development stage as opposed to customization/fine-tuning, where our work provides tools and addresses considerations for **model users** who would like to leverage an existing pre-trained model and customize the model given a limited budget.
>
> What is conducted in this work is more similar to fine-tuning but in an unsupervised/self-supervised manner. We think this idea of selecting pre-fine-tuning data from raw data (rather than curated labeled datasets) is relatively new, and being able to establish a visible performance margin with such a small budget is a novel result.
>
> We thank the reviewer for the comment. We are still searching for a better term and considering all the options.

---

> ### Author Response · Authors · 2023-11-21
> **Miscellaneous-1: $D_S$, $D_R$, etc. are used interchangeably to represent (finite) datasets and distributions**
>
> **Miscellaneous-1: It seems sometimes $D_S$, $D_R$, etc. are used interchangeably to represent (finite) datasets and distributions. Perhaps the authors can make this distinction clearer?**
>
> **Re.:** We agree with the reviewer's suggestion. This is a slight abuse of notation that is introduced on Page 4 before Eq. (2). The intention is to avoid introducing additional notations for the ease of general readers. But it does come at a cost of mathematical clarity.
>
> We are currently working on simplifying the notation systems to make it more direct and intuitive. We will keep this in mind and use separate notations for empirical distributions and sets of samples in an updated version.

---

> ### Author Response · Authors · 2023-11-21
> **Miscellaneous-2: last term of the expansion in Eq. (3)**
>
> **Miscellaneous-2: In Eq. (3), should the last term of the expansion include $\lambda(D_U-D_S)$ instead of just $\lambda D_U?$**
>
> **Re.:** We apologize for the potential unclarity here. For this equation that is currently in Appendix B.2, consider $OT (\lambda D_U+(1-\lambda) D_S, D_R)$ as a function in its first input–namely, $f(\lambda D_U+(1-\lambda) D_S)$.
>
> For $D_U$ being the small set of new data added during fine-tuning, it can be considered a small perturbation term. It is clearer if we replace $D_S$ with $x_0$ and $D_U$ with $\Delta x$, which gives
>
> $f(\lambda D_U+(1-\lambda) D_S) := f((1-\lambda)x_0 + \lambda \Delta x) \approx f(x_0 + \lambda \Delta x)$
>
> $= f(x_0) + f'(x_0)  [(x_0 + \lambda \Delta x) - x_0]$
>
> $=f(x_0) + f'(x_0)  \lambda \Delta x$
>
> $D_U$ (the $ \Delta x$ here) is already the increment term so we don't need the difference $(D_U-D_S)$.

---

> ### Author Response · Authors · 2023-11-21
> **Miscellaneous-3: The version of TAPT here seems to be the "curated-TAPT"**
>
> **Miscellaneous-3: The version of TAPT here seems to be the "curated-TAPT" described in Gururangan et al. rather than "TAPT", which I've more typically seen as just training on the unlabeled end-task inputs.**
>
> **Re.:** Thanks for the catch. The authors did overlook distinguishing the variations of this approach. Yes, what is being referred to in this manuscript should be addressed as "curated-TAPT". This is important and thanks for bringing it up. **We have changed all the references of TAPT to curated-TAPT (TAPT with a curated dataset, TAPT/c).**

---

> ### Author Response · Authors · 2023-11-21
> **Miscellaneous-4: In Table 3, it seems TAPT should be bolded for RCT.**
>
> **Miscellaneous-4: In Table 3, it seems TAPT should be bolded for RCT. Also the text right above mentions "All domains" but it seems to be missing from the table?**
>
> **Re.:** Thanks for the catch. The authors apologize for the typos/inconsistencies. The comment on the performance of "All domains" was misplaced during editing and was intended for Table 2. We have fixed these parts and also checked other contents.

---

> ### Author Response · Authors · 2023-11-22
> **Rebuttal period ending–we anticipate your feedback!**
>
> Dear Reviewer cFNr,
>
> As the rebuttal/author discussion period is closing, we sincerely look forward to your feedback. The authors are deeply appreciative of your valuable time and efforts spent reviewing this paper and helping us improve it.
>
> It would be very much appreciated if you could once again help review our responses and additional results and let us know if these address or partially address your concerns and if our explanations are heading in the right direction.
>
> Please also let us know if there are any further questions or comments about this paper. We strive to consistently improve the paper and it would be our pleasure to have your precious feedback!
>
> Kind Regards,\
> Authors of Submission530

---

> > ### Comment · Reviewer_cFNr · 2023-11-22
> >
> > Thanks to the authors for their receptiveness to feedback and detailed responses! Overall, they provide very useful explanations for better contextualizing the paper. Given the new additions, I do have some further questions/comments:
> >
> > **Presentation**
> > - **(Pre-fine-tuning vs continued pre-training)** I think model users v.s. pre-trained model providers is a useful motivator (though the lines could still be a bit blurry depending on who the model users are)? Also, a bit nitpicky, but what about (non-curated) TAPT, which involves a user's specific end-task yet was originally proposed under continued pre-training?
> > - **(Zero-shot settings)** I'd suggest making it clear earlier that targeted fine-tuning is optional in some cases. Much of the exposition before the experiments seems to assume there will be two fine-tuning stages, so it is a bit surprising when zero-shot settings come up in the experiments.
> > - **(Current/Future Edits)** I think the current edits are definitely steps in the right direction, but it does seem like there is still a decent amount of non-trivial re-organizing to do as the authors mention in their reply.
> >
> > **Experiments**
> > - **(Ablations)** I appreciate the author's explanations for not including these ablations and understand why they are not trivial to run. However, I still think such experiments are worth the effort in order to properly evaluate what i take to be the main idea from the paper, which is that we should avoid redundancy with pre-training.
> >   - **(Feature spaces)** I appreciate the new results in Table 17, which show robustness to choice of pre-trained representation. But I think this still leaves open the question as to whether access to pre-trained features itself is a confounder, precisely because of the invariances it may encode. To my understanding, only GOT-D makes use of pre-trained features for selection, so it would be nice to have a baseline that uses the same features + more standard distribution matching: ideally this would be a modified DSIR, but another possibility is to train a discriminative classifier on pre-trained features that predicts whether an example comes from the target distribution.
> >   - **(Ablations of pre-processing)** Generally, I think the candidate pool should be controlled between data selection methods as much as possible. Currently the pre-processing for GOT-D is not mentioned as a core aspect of the method (i.e. only appearing in Appendix) but it inherently is another form of data selection (that could in principle be combined with an approach like DSIR).
> > - **(Significance)** I think the extra context here is helpful re: what gains other baselines achieve at larger budgets. My broad impression is that maybe these same continued pre-training benchmarks are not the best settings to highlight the significance of GOT-D in the pre-fine-tuning regime (given that improvements are small relative to error bars)? I would be very curious to see what happens as you reach the budgets of continued pre-training or on more specialized/harder tasks?

---

> ### Author Response · Authors · 2023-11-23
> **Further response on Presentation**
>
> The authors thank Reviewer cFNr for the timely and detailed response. This insightful discussion has been very constructive to the development of this work and is deeply encouraging for the authors ☺️.
>
> **(Pre-fine-tuning vs continued pre-training)**
>
> The distinction between “model builders” and “model users” is commonly seen in the literature related to data selection problems in <data acquisition> or <data market> scenarios, where they define parties of different interests to facilitate discussion on data transaction and data economic problems. These concepts are natural and practically relevant, so we borrowed them into the development of this work. With the widespread adaptation of pre-trained LLMs and the increasing ability for model customizations (as in OpenAI's recent announcements), we think this setup would be timely and favorable.
>
> (non-curated) TAPT was proposed in an earlier stage when the wide adoption of LLMs was still out of reach for most. Since most researchers were working on model development, such distinctions were not as emphasized. From today's perspective, the builders of pre-trained LLMs are likely different from the end users, where the former might not have access to information about downstream tasks and the latter might not have the capability to continue pre-training the model on a large scale. We think it is more likely that TAPT (curated or non-curated version) will be used by end model users instead of pre-training model builders.
>
> **(Zero-shot settings)**
>
> We agree with the reviewer’s suggestion. We are adding the description that
>
> > The pre-fine-tuned model can be directly deployed to downstream tasks without further fine-tuning.
>
> And make it clear that the second fine-tuning stage is optional. This should make the presentation more coherent and natural.
>
> **(Current/Future Edits)**
>
> Thanks for reviewing our updated manuscript. We have completed all the technical revisions and additional experiments. We are taking the time to fully polish up the presentation for better readability in an effort to make it more accessible to a broader audience that is not necessarily experts on data selection problems or LLMs.

---

> ### Author Response · Authors · 2023-11-23
> **Further response on Experiments**
>
> **(Ablations)**
>
> We fully understand the reviewer’s concern about the differences in design choices between the methods being compared with. Indeed, the current comparison with DSIR has multiple differences. *We conducted better-controlled ablation studies with TAPT.*
>
> DSIR is based on the idea of importance resampling, using the weight of a generative model to help determine whether a sample is from the candidate distribution or target distribution and selecting the highest-scoring ones. As the original paper shows, this *reduces* the KL-divergence of the selected samples to the target domain. Yet, it does not guarantee that the select data to *minimize* this distributional distance. Additonal evaluations are required to understand to what extent samples selected by DSIR *reduce* KL-divergence or match the target distribution. (As a comparison, [GIO] selects data by explicitly minimizing the KL-divergence between selected data and the target data. However, the implementation of this idea is extremely costly and involves using a cluster of large servers for selecting data from a million-scale dataset, which most cannot afford.)
>
> > [GIO] Dante Everaert and Christopher Potts. Gio: Gradient information optimization for training dataset selection. arXiv preprint arXiv:2306.11670, 2023.
>
> **(Feature spaces and pre-processing)**
>
> The KNN-based TAPT may be a better fit for comparing the idea of *selecting samples closest to the target* vs. *selecting close but underrepresented samples (i.e., avoiding redundancy)*. We conducted direct comparisons with TAPT to showcase the effect of avoiding redundant samples
>
> In all the experiments in this work, TAPT uses a pre-trained feature embedder. In the BERT experiments, TAPT uses the VAMPIRE embedder pre-trained on the domain dataset as suggested in the original paper for these tasks. *In the rebuttal, TAPT uses <sentence-transformer> for embedding, which is also evaluated with GOT-D.* As discussed, in Fig.5, the selection TAPT well matches the target distribution but also has a high overlapping with the pre-training domain. As can be seen from Table 17, when implementing GOT-D with sentence-transfomer, GOT-D still outperforms TAPT by a visible margin. This difference can only be attributed to the philosophy of selecting the closest samples vs. selecting close but underrepresented samples (i.e., avoiding redundancy).
>
> Also, throughout this work, in GPT-2 experiments where the data is from a single domain (OWTC), all data selection methods select from the same pool of candidates without additional pre-processing.  For experiments involving multiple domains, GOT-D shares mostly the same pre-processing as DAPT/TAPT, essentially refining the scope of selection to the closest domain. As the reviewer suggests, this process can be considered a data selection method by itself (which is similar to DAPT) that can work in combination with DSIR. Our intention for designing the experiments in its current form is that we consider it an advantage of DSIR that it is able to select from all domains simultaneously thanks to its highly optimized scalability. We expect that should bring it an edge compared to other methods that select from a confined domain.
>
> Thus, a more controlled, fair comparison was conducted between TAPT and GOT-D, which more sharply concentrates on comparing the idea of *selecting samples closest to the target* vs. *selecting close but underrepresented samples (i.e., avoiding redundancy)*.
>
> We hope these results could help the reviewer to strengthen the confidence in supporting the idea of this work 🙂.
>
> **(significance)**
>
> Admittedly, these BERT results are a bit unconventional as their performance improvements are not much higher even in continued pre-training papers. We chose these experiments as they are better *contextualized* by previous works and could help readers make connections. At least for domain-specific tasks (Sec. 3.3), we think the averaged performance sees a clear trend and distincts the proposed methods from other baselines. GLUE experiments in (Sec. 3.4) are provided as an extended evaluation as the performance on these tasks is not meant to be easily improved by design.
>
> Moreover, this is also our motivation to include diverse experiments beyond these, such as de-toxification and zero-shot experiments with larger GPT models. *The results are these tasks are substantial*, which should provide evidence for the promising potential of the proposed approach.
>
> The authors would like to note that this work already includes a non-trivial amount of experiments in both quantity and resources involved. The team is definitely interested in extending this idea into constructing pre-training data for LLMs. Given that training a contemporary language model effectively and efficiently on a large scale remains a challenging research problem at this moment, we think that exploration could be an independent work by itself 🙂.

---

> > ### Author Response · Authors · 2023-11-23
> >
> > Dear Reviewer cFNr,
> >
> > The authors once again express their appreciation for the reviewer's effort and dedication in helping with the improvement of this work. We enjoyed the insightful and constructive exchange and believe the development of this work as well as our understanding of it has benefited substantially from the process.
> >
> > If the rebuttal and our responses have addressed or partially addressed your questions and concerns, or if revisions to the manuscript and additional results have improved the quality of this work, the authors would greatly appreciate it if the reviewer would consider raising the score in support of publishing this work (even if not in full support 🙂).  Your review and feedback are greatly appreciated by all of the authors.
> >
> > Besides, here are the authors' thoughts on this work that were not explicited in the manuscript.
> >
> > This work contributed a principled approach that is substantially novel and timely. The idea of using the free gradient of Optimal Transport for fine-tuning pre-trained models offers a new perspective in selecting data with high efficacy and at the same time tackles the plaguing issue of computational overhead. With the widespread adoption of LLMs in end scenarios and increasing ability for model customization, the research on data selection for fine-tuning LLMs is trending and attracting growing interest from broad audiences. Well-grounded in both theories and empirical insights, this work holds the promising potential to provide methodological inspiration to a field currently packed with overly heuristic methods and ad-hoc designs. With the revised methodology section with substantial clarity and rigor, the authors believe the intellectual merit of this work would be an asset for facilitating innovations in the research field.
> >
> > Though poised for LLMs, this principled approach essentially applies to fine-tuning any large pre-trained models, such as Vision Transformer (ViT), multimodal models (e.g., CLIP), or Automatic Speech Recogntion (ASR), etc., without needing additional customization. Publication of this work would boost dissemination the intellectual contributions and facilitate subsequent works, holding broad potential to benefit a variety of fields and provide conceptual inspiration to large audiences.
> >
> > Thanks again,\
> > Authors of Submission530

---

### Official Review · Reviewer_2m8a · 2023-10-31

**Soundness:** 2 fair
**Presentation:** 3 good
**Contribution:** 2 fair
**Rating:** 3
**Confidence:** 4

**Summary:**

This paper studies data selection for pre-fine tuning a pre-trained language model for fine-tuning.  The proposed approach uses OT to select a budgeted amount of data from a pre-training corpus that is different from the original pre-training data and similar to the downstream task data  to first continue training, then fine-tune on new data for the downstream task.  The proposed approach GOT-D shows improvements on domain adaptation tasks, GLUE benchmark, and fairness evaluations.

**Strengths:**

The proposed approach has clear motivation and intuition as the pre-training dataset can be quite different from the training set for the downstream tasks.  Holding out a specific subset based on downstream evaluations can help gradual transfer of the model leading to better performance.  OT is also a natural choice to compare distributions and select data, and authors note that this can be done quickly.   Authors compare with recent prior work in the space and demonstrate better performance on all tasks.

**Weaknesses:**

* The proposed approach shows only marginal improvement on a number of domain adaptation and GLUE benchmarks - les than 1% over prior approaches, and in some cases only 1% higher than baseline performance.  Another concern with the evaluation is that it is only evaluated on small models (128M) that may be trained with more limited data.  In contrast, it would be beneficial to evaluate whether using the pre-fine tune set also leads to better performance on a larger model (1/7B+).

* While the performance on toxic detection are strong, these improvements come at the cost of increased perplexity.  Further the (token?) perplexity of these models are quite high.  Regarding the above concern, authors can evaluate on a larger model with reduced perplexity to see if the impact increases perplexity at the cost of decreasing toxicity.

* Authors only conduct limited evaluation on the pre-fine-tuning dataset size at 50K and 150K.  An experiment at different amounts of data from small data to continual pre-training with comparison to prior works can help to better motivate GOT-D as an approach for selecting pre-fine tuning data.

* Part of the motivation appeared to be not needing as much data for the downstream task, however it appears they still need the full fine-tuning data.

**Questions:**

Q1: Where does the data for pre-fine-tuning come from? In the main text, the authors suggest that it can come from the Pile, however GPT-2 is trained on WebText, and not necessarily subsets of the Pile.  in this case, the experiments are adding more pre-training data where even in small amounts are expected to increase performance.

Q2: What type of data is selected from the pre-training set? Is there a comparison selected data vs. fine-tune data and random pre-train data?

---

> ### Author Response · Authors · 2023-11-21
> **Weakness-1a: Performance improvements in Section 3.3/3.4 seem marginal**
>
> **Weakness-1a: Performance improvements in Section 3.3/3.4 seem marginal.**
>
> **Re.:** The authors did realize this presentation issue not long after submitting the manuscript.
>
> 8 NLU tasks with a pre-defined domain (DAPT tasks, Section 3.3/Table 2, 3) are first presented in DAPT paper, where they show continued pre-training RoBERTa model for 2\~8B tokens on a target domain (equivalent to **20M\~80M** samples with 100-token length) could improve the performance on the tasks in this domain by **2.3%** on average (the improvement is less than 2% on 5 tasks). DSIR achieves an average of **1.56% improvements after continued pre-training on 25M samples.**
>
> In this work, we show that favorable performance improvements can be established with a **much smaller** fine-tuning budget if the data is selected appropriately. We achieved an average of 1.13% improvement after fine-tuning with only 150k samples **(~0.5% of the data budget for continued pre-training) in Table 2 and a 0.76% improvement with 50k samples in low-resource scenarios in Table 3. It can be seen from the tables that the rate of performance improvements for GOT-D significantly **outpaces** that of baseline methods, highlighting the efficacy and data efficiency of the proposed method.
>
> The results for 8 NLU tasks without a pre-defined domain (GLUE tasks Section 3.4/Table 4) are meant to be interpreted in a slightly different way. GLUE tasks are designed to measure the **general capability** of a language model. These tasks are specifically selected to measure very different aspects of the language model and it is hard to achieve a high score in all aspects at the same time.
>
> We conducted this experiment as a test to see whether light fine-tuning can improve the scores on this kind of benchmark. DSIR achieves a **2%** improvement in GLUE score by curating pre-training data and training a BERT model from scratch for **51.2M** samples. **It is expected that performance improvements can be difficult.** As mentioned in the discussion in the paper, the proposed light pre-fine-tuning is more beneficial for tasks that benefit from certain task-relevant knowledge, rather than the ones relying on the fundamental capability of the model. It is an interesting result for us to see that light pre-fine-tuning with the proposed method achieves more than 1% performance improvement on the GLUE benchmark.
>
> It is our responsibility not being able to provide sufficient discussion after the experiments to elaborate on the **takeaway** for each result.
>
> **Besides, we have added a new set of experiments for pre-fine-tuning GPT models for zero-shot classification tasks, where the performance improvements of our method are 6.6%~13.9% after pre-fine-tuning with 40k samples.**

---

> > ### Comment · Reviewer_2m8a · 2023-11-22
> > **Reviewer response to author clarification on Weakness-1a**
> >
> > Can I confirm that in Tables 2-4, the data selection only applies to GOT-D? For example in Table 2, GOT-D is only pre-fine tuning on 150K samples whereas the other approaches: DAPT and DSIR are training on more data? If so, I think the wording in the paper should be changed as it reads "DAPT (Gururangan et al., 2020) and DSIR (Xie et al., 2023), sharing the same selection budget as GOT-D for fair comparison."

---

> ### Author Response · Authors · 2023-11-21
> **Weakness-1b: It is only evaluated on small models (128M)...would be beneficial to evaluate... on a larger model (1/7B+).**
>
> **Weakness-1b: It is only evaluated on small models (128M)...would be beneficial to evaluate... on a larger model (1/7B+).**
>
> **Re.:** Thanks for the comment. In light of the review, we conducted additional experiments with larger models on zero-shot classification tasks. **We evaluate OpenAI's GPT-2 XL (1.5B) and Eleuther AI's GPT-neo (2.7B)**, which are widely used in zero-shot learning research. Our analysis encompasses two benchmark tasks: AG News, a text classification challenge focusing on news categorization, and BoolQ, a question-answering dataset involving natural yes/no questions.
>
> For GPT-2 XL whose pre-training data is from a single domain–OpenWebTextCorpus(OWTC), **we use the same data as the candidate dataset**. This setting is the same as the NLG task in Section 3.2. GPT-neo is pre-trained on The Pile dataset. We construct a substitute candidate dataset with samples from 7 domains. This setting is the same as NLU tasks in Section 3.3/3.4. The experiment procedure is similar to the NLG task in Section 3.2. Given a few thousand **unlabeled** training samples (5K for AG News and 9K for BoolQ) as the target data, we test different data selection methods (GOT-D, DSIR, curated-TAPT (TAPT with a curated dataset, TAPT/c)) select samples from the candidate dataset to fine-tune the model. Then, we test the zero-shot classification accuracy of the fine-tuned model on target tasks and measure the performance improvements gained from each data selection method.
>
> These LLM experiments take significant resources. A sizeable number of models need to be fine-tuned considering the number of tasks, baselines, data sizes, repetitions, and many ablations for design choices.
>
> **The performance improvements of our method are 6.6%~13.9% after pre-fine-tuning with 40k samples, visibly outperforming baseline methods. These favorable results show the consistent efficacy and data efficiency of the proposed approach.**

---

> > ### Comment · Reviewer_2m8a · 2023-11-22
> > **Reviewer response to author clarification on Weakness-1b**
> >
> > Thank you for running additional experiments on the zero-shot tasks.  The results here are promising, however I have two concerns:
> >
> > 1) How are these datasets chosen? There are many zero-shot tasks the authors can select and are normally used beyond the two suggested such as HellaSwag, NLU, ARC, etc.
> >
> > 2) I am concerned with whether this is a fair/practical evaluation.  The purpose of the zero-shot evaluation is to test model capability to produce correct answers.  Selecting a corpus based on the questions from the downstream task shifts the task significantly.   One could easily produce a very similar answer from the training set, and from this, the model only needs to memorize the samples seen in the fine-tuning set.  I'm concerned that GOT-D is selecting passages from the training set that contain the question/answer for fine-tuning.

---

> ### Author Response · Authors · 2023-11-21
> **Weakness-1c:  these improvements on toxic detection come at the cost of increased perplexity**
>
> **Weakness-1c: While the performance on toxic detection are strong, these improvements come at the cost of increased perplexity...evaluate on a larger model with reduced perplexity...**
>
> **Re.:** Thanks for the feedback. The authors would like to first clarify a potential misperception.
>
> In the model detoxification task in Section 3.2, pre-fine-tuning data is selected from the candidate dataset that is the **same as the model (GPT-2)'s pretraining domain**–OpenWebTextCorpus(OWTC). As a result, the fine-tuned model's perplexity (ppl) on the pre-training domain actually **decreases** compared to without pre-fine-tuning. In particular, our method GOT-D_constrast, which selects samples that pull the model away from the distribution of toxic examples, achieves **one of the best perplexity reductions** after pre-fine-tuning while effectively reducing the model's toxicity level.
>
> The ppl scores are inherent for GPT-2, which might appear higher compared with state-of-the-art LLMs. In light of the reviewer's suggestion, we also conducted experiments on larger models–GPT-2 XL (1.5B) and GPT-neo (2.7B). The proposed pre-fine-tuning method improves the model's utility on zero-shot classification tasks.

---

> > ### Comment · Reviewer_2m8a · 2023-11-22
> > **Reviewer response  to author rebuttal on weakness-1c**
> >
> > Thank you for providing additional detail on the toxicity experiments.  The PPL values still appear high for a model which is trained on OWT and evaluated on OWT.  I would expect loss around 3.1 which should lead to PPL in the low-mid 20s (see https://github.com/karpathy/nanoGPT for example).  The GOT-D approach looks to have perplexity above 30, which is more importantly  higher than other approaches in Table 1.

---

> ### Author Response · Authors · 2023-11-21
> **Weakness-2:  experiment at different amounts of data from small data to continual pre-training**
>
> **Weakness-2: An experiment at different amounts of data from small data to continual pre-training with comparison to prior works can help to better motivate GOT-D as an approach for selecting pre-fine tuning data.**
>
> **Re.:** The authors agree that the experiment mentioned by the reviewer would be desirable. Yet these LLM experiments take significant resources and can be considerably costly. For example, fine-tuning OpenAI's davinci-002 on a set of **100K short samples with a max length of 128 tokens using recommended settings with OpenAI's API costs $1,500.** A sizeable number of models need to be fine-tuned considering the number of tasks, baselines, data sizes, repetitions, and many ablations for design choices. Continued pre-training experiments are very difficult to conduct with an academic budget.
>
> In light of the direction pointed out by the reviewer, we conducted **ablation studies on pre-fine-tuning data sizes** with our available resources. The results are presented in Table 16 at the end of Appendix. Performance improvements appear to comply with a **log-linear scaling law relationship** with pre-fine-tuning data size with the performance recording constant improvement every time the fine-tuning data budget is doubled.

---

> ### Author Response · Authors · 2023-11-21
> **Weakness-3: it appears still need the full fine-tuning data**
>
> **Weakness-3: Part of the motivation appeared to be not needing as much data for the downstream task, however it appears they still need the full fine-tuning data.**
>
> **Re.:** As the title of our paper suggests (“get more for less”), one of the major motivations for the proposed data selection/pre-fine-tuning scheme is it is able to provide performance improvements on target tasks **for free**. We show in the experiments that with the same labeled fine-tuning data, pre-fine-tuning on data selected by the proposed method is able to provide **favorable performance gain** that is comparable with the level associated with continued pre-training on orders of magnitudes more data.
>
> This showcases the importance of proper data curation for language models and sheds light on a promising direction. **The pre-fine-tuned model can be directly deployed to downstream tasks without further fine-tuning.** For example, as in the additional experiments (and also the NLG task in Section 3.2), in zero-shot tasks where there are no labeled training data, this provides a viable approach to improving performance on the target tasks and yields concrete benefits.

---

> ### Author Response · Authors · 2023-11-21
> **Question-1: GPT-2 is trained on WebText, and not necessarily subsets of the Pile**
>
> **Question-1: Where does the data for pre-fine-tuning come from? ...GPT-2 is trained on WebText, and not necessarily subsets of the Pile.**
>
> **Re.:** Thanks for providing the feedback. The authors acknowledge the responsibility for causing the potential misperception.
>
> Due to the page limit of the main paper, the details of models/candidate datasets for each experiment are summarized in Appendix C.1/C.2. For GPT-2 base/XL that are pre-trained on a single domain–OpenWebTextCorpus(OWTC), **we use the same data as the candidate dataset for the NLG task in Section 3.2 as well as the additional experiments.** All baseline methods select from the same pool of candidate data. The setting is well-aligned and consistent with assumptions in the development of our method.
>
> Thus, the selected data for pre-fine-tuning does not add anything new to the scope of the model's pre-training domain. The **performance gain** achieved on the downstream tasks can **reasonably support** the idea that "pre-fine-tuning with properly selected data better prepares the model for the downstream task."
>
> Our replica of The Pile dataset consisting of data from 7 domains is used for NLU tasks in Section 3.3/3.4. with BERT as well as the additional experiments with GPT-neo, which is pre-trained on The Pile dataset.

---

> ### Author Response · Authors · 2023-11-21
> **Question-2: What type of data is selected from the pre-training set?**
>
> **Question-2: What type of data is selected from the pre-training set? Is there a comparison selected data vs. fine-tune data and random pre-train data?**
>
> **Re.:** The authors agree that it will be desirable to direct compare the samples selected by each method and random selection from pre-training data.
>
> In the additional experiments with the “AG News” task where all the settings are best aligned, we **conduct visualization** of samples selected by each method and the distribution of target task data and pretraining/candidate data, offering insights on how the proposed method works differently from the distribution-matching approaches. As shown in Figure 16, the comparison shows a clear contrast. **Both DSIR and TAPT/c select samples that match the distribution of the target task data which has a high overlapping with the distribution of the candidate data** where the model is already pre-trained on. Thus, with such a small data budget, the information gain provided from pre-fine-tuning on these samples is naturally marginal. In contrast, GOT-D selects predominately formal business news which is **highly underrepresented in the candidate dataset but important for the target task**. Pre-fine-tuning the model with these samples provides **a more direct benefit** in aligning the model with the target tasks, effectively validating the idea of this work.

---

> > ### Comment · Reviewer_2m8a · 2023-11-22
> > **Reviewer response to author clarification on Question 2**
> >
> > Thank you for running this additional experiment and providing additional details on the data selection.  It is interesting that GOT-D picks very different samples compared with DSIR and TAPT.

---

> ### Author Response · Authors · 2023-11-22
> **Rebuttal period ending–we anticipate your feedback!**
>
> Dear Reviewer 2m8a,
>
> As the rebuttal/author discussion period is closing, we sincerely look forward to your feedback. The authors are deeply appreciative of your valuable time and efforts spent reviewing this paper and helping us improve it.
>
> It would be very much appreciated if you could once again help review our responses and additional results and let us know if these address or partially address your concerns and if our explanations are heading in the right direction.
>
> Please also let us know if there are any further questions or comments about this paper. We strive to consistently improve the paper and it would be our pleasure to have your precious feedback!
>
> Kind Regards,\
> Authors of Submission530

---

> ### Author Response · Authors · 2023-11-23
> **Further response to Weakness-1a**
>
> > "Can I confirm that in Tables 2-4, the data selection only applies to GOT-D? For example in Table 2, GOT-D is only pre-fine tuning on 150K samples whereas the other approaches: DAPT and DSIR are training on more data? If so, I think the wording in the paper should be changed as it reads "DAPT (Gururangan et al., 2020) and DSIR (Xie et al., 2023), sharing the same selection budget as GOT-D for fair comparison."
>
> We are sorry for causing the potential confusion. All the numbers reported in this work are experiments conducted by our own team and at the same scale of GOT-D (150K). The numbers provided in the rebuttal
>
> > "8 NLU tasks with a pre-defined domain (DAPT tasks, Section 3.3/Table 2, 3) are first presented in DAPT paper, where they show continued pre-training RoBERTa model for 2\~8B tokens on a target domain (equivalent to 20M\~80M samples with 100-token length) could improve the performance on the tasks in this domain by 2.3% on average (the improvement is less than 2% on 5 tasks). DSIR achieves an average of 1.56% improvements after continued pre-training on 25M samples."
>
> are cited from the original works  "DAPT (Gururangan et al., 2020) and DSIR (Xie et al., 2023)", providing references on the scale of performance improvements on these tasks to demonstrate the significance of results achieved in this paper.

---

> ### Author Response · Authors · 2023-11-23
> **Further response to weakness-1c**
>
> > "Thank you for providing additional detail on the toxicity experiments. The PPL values still appear high for a model which is trained on OWT and evaluated on OWT. I would expect loss around 3.1 which should lead to PPL in the low-mid 20s (see https://github.com/karpathy/nanoGPT for example). The GOT-D approach looks to have perplexity above 30, which is more importantly higher than other approaches in Table 1."
>
> Thanks to the reviewer for providing the reference and sharing with us your experiences with these models. For the higher base ppl, this is limited by the capability of GPT-2 models. We use the official version from huggingface that is pretrained by OpenAI without modification. The state-of-the-art large models such as GPT-4/LlaMA-2 are already pre-treated on these toxicity benchmarks before the release such that we cannot get a meaningful score when evaluating on these benchmarks.
>
> We fully understand the reviewer's concern. It is important to control the degradation of the utility of the model while aligning its output. *Even for the state-of-the-art models, alignment always comes with a visible cost to the model's utility as the alignment process is essentially guiding the model to not output the most natural continuation it learned from the pre-training domain.*
>
> For the comparison in Table 1, since the model is pre-trained on OWTC and tested the ppl on OWTC, if the alignment process is *ineffective* and we merely continue training the model on general OWTC samples,  we will see a reduction in ppl on OWTC as this is exactly the training objective. Thus, the effort in the alignment process is to achieve an effective reduction in toxicity level while controlling the loss in utility of the model–the loss is anticipated.
>
> In the results presented in Table 1, all of our methods substantially decreased the toxicity level in the model output, did not raise the ppl on OWTC, and well-preserved the utility of the model. **Especially for GOT-D_contrast (ours), it achieves one of the best toxicity reduction performances while also outperforming other methods in preserving the utility of the model.** This shows our methods not only effectively carry out their duty, but also achieve a better Pareto optimality for the tradeoff between model utility, which is favorable success 🙂.

---

> ### Author Response · Authors · 2023-11-23
> **Further response to Weakness-1b**
>
> > "Thank you for running additional experiments on the zero-shot tasks. The results here are promising, however I have two concerns:
> How are these datasets chosen? There are many zero-shot tasks the authors can select and are normally used beyond the two suggested such as HellaSwag, NLU, ARC, etc."
>
> Thanks for the suggestions. We chose AG News and BoolQ as they are *among the most commonly used tasks for evaluating zero-shot performance for GPT-2 models*. We think the reviewer's suggestions are also great options. Given the similarity in the format of these tasks, we have confidence that the proposed methods would also achieve favorable results on these tasks.
>
> > Ref: Zero-shot Text Classification With Generative Language Models, 2019
>
> > Ref: Calibrate Before Use: Improving Few-Shot Performance of Language Models, 2021
>
> ---
>
> > "I am concerned with whether this is a fair/practical evaluation. The purpose of the zero-shot evaluation is to test model capability to produce correct answers. Selecting a corpus based on the questions from the downstream task shifts the task significantly. One could easily produce a very similar answer from the training set, and from this, the model only needs to memorize the samples seen in the fine-tuning set. I'm concerned that GOT-D is selecting passages from the training set that contain the question/answer for fine-tuning."
>
> The reviewer's concern is definitely valid, as pronounced by this article
>
> > Ref: Pretraining on the Test Set Is All You Need, 2023
>
> This is exactly the reason why in GPT-2 experiments, we consider the target task data as **validation data** that can only be used to guide the selection of fine-tuning data *but never to be trained on the model*.
> AG News and BoolQ are NLU classification tasks. We used **unlabeled data disjoint from the test** as the validation for selecting data. *The model was never shown the question or the answer such that the concern for data contamination is avoided.*
>
> As shown in Table 15, these tasks of the following format
>
> | **Task** | **Prompt** | **Label Names** |   |   |
> |----------|------------|-----------------|---|---|
> | AGNews   |  Wall St. Bears Claw Back Into the Black (Reuters) Reuters - Short-sellers, Wall Street's dwindling band of ultra-cynics, are seeing green again.  |  World, Sports, **Business**, Science/Technology  	|   |   |
> | BoolQ	| New York state law does not require a license to own or possess long guns, but does require a permit to legally possess or own a pistol. However, all firearms must comply with the NY SAFE Act, which bans guns considered ``assault weapons'' from ownership by private citizens, unless they were owned prior to the ban. [Question: is it legal to carry a gun in nyc?  The answer is] 	|   **Yes**, No    	|   |   |
> |      	|        	|             	|   |   |
>
> We only use the text part (i.e., the paragraph) as the target for data selection *without the question or the answer*. The intention of "pre-fine-tuning" is to improve the model's familiarity with the domain of these texts by reducing its ppl. Then, the pre-fine-tuned model would have a better understanding ability of these domains to help with zero-shot QAs. Similarly, if we are going to select data for pre-fine-tuning the model for HellaSwag, we will only use the prompt part as the target to find data and not include the completions, desipte samples are disjoint from test.
>
> We hope this could help address the concern of the reviewer and help strengthen the confidence in supporting this work 🙂.

---

> > ### Author Response · Authors · 2023-11-23
> >
> > Dear Reviewer 2m8a,
> >
> > The authors once again express their appreciation for the reviewer's effort and dedication in helping with the improvement of this work. We enjoyed the insightful and constructive exchange and believe the development of this work as well as our understanding of it has benefited substantially from the process.
> >
> > If the rebuttal and our responses have addressed or partially addressed your questions and concerns, or if revisions to the manuscript and additional results have improved the quality of this work, the authors would greatly appreciate it if the reviewer would consider raising the score in support of publishing this work (even if not in full support 🙂).  Your review and feedback are greatly appreciated by all of the authors.
> >
> > Besides, here are the authors' thoughts on this work that were not explicited in the manuscript.
> >
> > This work contributed a principled approach that is substantially novel and timely. The idea of using the free gradient of Optimal Transport for fine-tuning pre-trained models offers a new perspective in selecting data with high efficacy and at the same time tackles the plaguing issue of computational overhead. With the widespread adoption of LLMs in end scenarios and increasing ability for model customization, the research on data selection for fine-tuning LLMs is trending and attracting growing interest from broad audiences. Well-grounded in both theories and empirical insights, this work holds the promising potential to provide methodological inspiration to a field currently packed with overly heuristic methods and ad-hoc designs. With the revised methodology section with substantial clarity and rigor, the authors believe the intellectual merit of this work would be an asset for facilitating innovations in the research field.
> >
> > Though poised for LLMs, this principled approach essentially applies to fine-tuning any large pre-trained models, such as Vision Transformer (ViT), multimodal models (e.g., CLIP), or Automatic Speech Recogntion (ASR), etc., without needing additional customization. Publication of this work would boost dissemination the intellectual contributions and facilitate subsequent works, holding broad potential to benefit a variety of fields and provide conceptual inspiration to large audiences.
> >
> > Thanks again,\
> > Authors of Submission530

---

### Official Review · Reviewer_GWmY · 2023-11-01

**Soundness:** 3 good
**Presentation:** 3 good
**Contribution:** 3 good
**Rating:** 6
**Confidence:** 3

**Summary:**

This paper introduces a data selection method GOT-D at the pre-fine-tuning stage to improve the task adaption performance. The setting assumes access to both the pre-training distribution and the target fine-tuning distribution, and the key idea is to prioritize samples that most effectively shift the pre-training distribution closer to the target fine-tuning distribution. GOT-D leverages the OT distance as the distribution discrepancy measure to construct the optimization objective. Experiments on diverse tasks demonstrate the efficacy of the proposed method.

**Strengths:**

1. This paper is well-motivated. It argues the data selection methods based on distribution matching do not directly apply to fine-tuning problems, then points out that the fine-tuned model reflects a weighted combination of pre-training distribution and fine-tuning distribution. This naturally introduces this paper’s idea that the selected data should pull the model toward the target task.

2. The evaluation spans a wide spectrum of tasks.

**Weaknesses:**

1. The baseline methods are limited. Although the related work discusses a batch of work for language data selection, only DSIR and DAPT (RTP on toxicity, TAPT on GLUE) are compared in the experiments. It will be more convincing to include more baseline methods or illustrate why some are excluded.

2. The performance improvements, particularly in Tables 3 and 4, seem marginal, especially when taking the standard deviations into account.

**Questions:**

1. To compute the OT distance, what’s the representation of the involved data points?

2. Because the pre-training dataset is undisclosed, the authors resort to a proxy dataset that has a composition proximate. Is it possible to implement a synthetic task where the pre-training dataset is known to verify the proposed method?

---

> ### Author Response · Authors · 2023-11-21
> **Weakness-1: more baseline methods or why some are excluded**
>
> **Weakness-1: It will be more convincing to include more baseline methods or illustrate why some are excluded.**
>
> > "...baseline methods are limited... only DSIR and DAPT (RTP on toxicity, TAPT on GLUE) are compared in the experiments... more convincing to include more baseline methods or illustrate why some are excluded."
>
> **Re.:** **We picked the most representative baselines. Most methods are not applicable to the scale of this problem.** Some others are either overly heuristic and ad-hoc or have an unaffordable computing overhead.
>
> Thanks for bringing up this interesting point! Most data selection methods discussed in the related work do not apply to the problem scale of this work. For example, data valuation/coreset methods often only scale to a few thousand samples. Some recent methods could potentially scale better at a cost of extreme computing overhead (e.g., [GIO] requires a cluster of servers for clustering) while the idea is essentially similar (matching distributions by explicitly minimizing KL-divergence.)
>
> We included the most representative baselines relevant to the task, covering the most important methodologies for this problem.
>
> - Distributional matching (DSIR)
> - Nearest neighbors (curated-TAPT (TAPT with a curated domain dataset, TAPT/c))
> - Curated domain dataset (DAPT)
> - External knowledge/model (RTP)
>
> To our knowledge, **this should cover existing principled approaches** toward this data selection problem.
>
> As far as the authors are aware, there also exists a number of heuristic methods for curating data for LLMs such as filtering data for quality (with some simple criteria), diversity, length, etc. These methods are inherently ad-hoc and overly empirical where the effectiveness highly depends on many design choices/threshold settings. Since these are not principled approaches and they mostly underperform in general cases (as evaluated in [DSIR]), we left out these methods in this paper.
>
> We appreciate this feedback from the reviewer and this discussion is helpful for revising our manuscript for a better presentation.
>
>
> > [GIO] Dante Everaert and Christopher Potts. Gio: Gradient information optimization for training dataset
> selection. arXiv preprint arXiv:2306.11670, 2023.

---

> ### Author Response · Authors · 2023-11-21
> **Weakness-2: Performance improvements, particularly in Tables 3 and 4, seem marginal.**
>
> **Weakness-2: Performance improvements, particularly in Tables 3 and 4, seem marginal.**
>
> > "... performance improvements, particularly in Tables 3 and 4, seem marginal, especially when taking the standard deviations into account"
>
> **Re.:** The authors did realize this presentation issue not long after submitting the manuscript.
>
> 8 NLU tasks with a pre-defined domain (DAPT tasks, Section 3.3/Table 2, 3) are first presented in DAPT paper, where they show continued pre-training RoBERTa model for 2\~8B tokens on a target domain (equivalent to **20M\~80M** samples with 100-token length) could improve the performance on the tasks in this domain by **2.3%** on average (the improvement is less than 2% on 5 tasks). DSIR achieves an average of **1.56% improvements after continued pre-training on 25M samples.**
>
> In this work, we show that favorable performance improvements can be established with a **much smaller** fine-tuning budget if the data is selected appropriately. We achieved an average of 1.13% improvement after fine-tuning with only 150k samples **(~0.5% of the data budget for continued pre-training) in Table 2 and a 0.76% improvement with 50k samples in low-resource scenarios in Table 3. It can be seen from the tables that the rate of performance improvements for GOT-D significantly **outpaces** that of baseline methods, highlighting the efficacy and data efficiency of the proposed method.
>
> The results for 8 NLU tasks without a pre-defined domain (GLUE tasks Section 3.4/Table 4) are meant to be interpreted in a slightly different way. GLUE tasks are designed to measure the **general capability** of a language model. These tasks are specifically selected to measure very different aspects of the language model and it is hard to achieve a high score in all aspects at the same time.
>
> We conducted this experiment as a test to see whether light fine-tuning can improve the scores on this kind of benchmark. DSIR achieves a **2%** improvement in GLUE score by curating pre-training data and training a BERT model from scratch for **51.2M** samples. **It is expected that performance improvements can be difficult.** As mentioned in the discussion in the paper, the proposed light pre-fine-tuning is more beneficial for tasks that benefit from certain task-relevant knowledge, rather than the ones relying on the fundamental capability of the model. It is an interesting result for us to see that light pre-fine-tuning with the proposed method achieves more than 1% performance improvement on the GLUE benchmark.
>
> It is our responsibility not being able to provide sufficient discussion after the experiments to elaborate on the **takeaway** for each result.
>
> **Besides, we have added a new set of experiments for pre-fine-tuning GPT models for zero-shot classification tasks, where the performance improvements of our method are 6.6%~13.9% after pre-fine-tuning with 40k samples.**

---

> ### Author Response · Authors · 2023-11-21
> **Question-1: To compute the OT distance, what’s the representation of the involved data points?**
>
> **Question-1: To compute the OT distance, what’s the representation of the involved data points?**
>
> **Re.:** In the original manuscript, OT is computed on embeddings from the **distilled-BERT** model.
>
> In the additional experiments, we ablate on the effect of choices of embedding space for computing OT distance. We tested embedding samples with **distilled-BERT, sentence-transformer, and BERT-tokens**. Lightweight and fast, the popular sentence-transformer uses a pre-trained all-MiniLM-L6-v2 model with 22M parameters as the backbone. It embeds up to 6 million samples/hours on a single GPU and is sometimes considered a 'default' option for sentence embedding in many NLP tasks. Token space isn't a proper embedding for OT (e.g., the distance on token space is not invariant to paraphrase). We are only listing it here for comparison.
>
> As can be seen from Table 16, the performance of sentence-transformer is mostly on par with distilled-BERT. **It suggests the choice of embedding space isn't a critical part of the data selection pipeline and any reasonable embedding space should work.**

---

> ### Author Response · Authors · 2023-11-21
> **Question-2: Is it possible to implement a synthetic task where the pre-training dataset is known to verify the proposed method?**
>
> **Question-2: Is it possible to implement a synthetic task where the pre-training dataset is known to verify the proposed method?**
>
> **Re.:**  GPT-2 is a perfect example as its training data is from a single domain–OpenWebTextCorpus(OWTC). In the NLG task in Section 3.2 and the additional experiments, **we use the same data as the candidate dataset** and all data selection methods (GOT-D, DSIR, TAPT/c) select from the same candidate dataset. This should provide a fair comparison. In both of these experiments, our methods demonstrate promising results.

---

> ### Author Response · Authors · 2023-11-22
> **Rebuttal period ending–we anticipate your feedback!**
>
> Dear Reviewer GWmY,
>
> As the rebuttal/author discussion period is closing, we sincerely look forward to your feedback. The authors are deeply appreciative of your valuable time and efforts spent reviewing this paper and helping us improve it.
>
> It would be very much appreciated if you could once again help review our responses and additional results and let us know if these address or partially address your concerns and if our explanations are heading in the right direction.
>
> Please also let us know if there are any further questions or comments about this paper. We strive to consistently improve the paper and it would be our pleasure to have your precious feedback!
>
> Kind Regards,\
> Authors of Submission530

---

> > ### Comment · Reviewer_GWmY · 2023-11-23
> >
> > I acknowledge and appreciate the response provided by the authors, which provides additional clarity and addresses my concerns/questions.

---

> ### Author Response · Authors · 2023-11-23
>
> Dear Reviewer GWmY,
>
> The authors once again express their appreciation for the reviewer's effort and dedication in helping with the improvement of this work. We enjoyed the insightful and constructive exchange and believe the development of this work as well as our understanding of it has benefited substantially from the process.
>
> If the rebuttal and our responses have addressed or partially addressed your questions and concerns, or if revisions to the manuscript and additional results have improved the quality of this work, the authors would greatly appreciate it if the reviewer would consider raising the score in support of publishing this work 🙂.  Your review and feedback are greatly appreciated by all of the authors.
>
> Besides, here are the authors' thoughts on this work that were not explicited in the manuscript.
>
> This work contributed a principled approach that is substantially novel and timely. The idea of using the free gradient of Optimal Transport for fine-tuning pre-trained models offers a new perspective in selecting data with high efficacy and at the same time tackles the plaguing issue of computational overhead. With the widespread adoption of LLMs in end scenarios and increasing ability for model customization, the research on data selection for fine-tuning LLMs is trending and attracting growing interest from broad audiences. Well-grounded in both theories and empirical insights, this work holds the promising potential to provide methodological inspiration to a field currently packed with overly heuristic methods and ad-hoc designs. With the revised methodology section with substantial clarity and rigor, the authors believe the intellectual merit of this work would be an asset for facilitating innovations in the research field.
>
> Though poised for LLMs, this principled approach essentially applies to fine-tuning any large pre-trained models, such as Vision Transformer (ViT), multimodal models (e.g., CLIP), or Automatic Speech Recogntion (ASR), etc., without needing additional customization. Publication of this work would boost dissemination the intellectual contributions and facilitate subsequent works, holding broad potential to benefit a variety of fields and provide conceptual inspiration to large audiences.
>
> Thanks again,\
> Authors of Submission530

---

### Author Response · Authors · 2023-11-21
**Summary response**

Thanks to all reviewers and AC for their dedicated work.

We are appreciative that all reviewers clearly understood the context and motivation of this work as well as the exact methodology of our proposed approach. This work is fortunate to receive all positive feedback on its methods and ideas.

All reviewers agree this work to be **well-motivated, clearly contextualized, the idea and development are quite natural/interesting/novel/principled, and includes extensive experiments.** Reviewers 2m8a, GWmY, nPTw comment on the proposed method to be **efficient and scalable, promising for scaling up to large datasets.**

The authors are grateful for the detailed issues pointed out by the reviewers. The questions mostly reside on **presentation issues (e.g., clarifications of experiment details, visualization of data selection results)**, and the need for **additional ablation studies on design choices**.

In light of the insightful reviews, we have conducted **a new set of experiments on larger models with controlled ablations on design choices**, **formalized the methodology section into rigorous lemmas/theorems with proofs/remarks**, **added visualization for insights into selected data**, **revised the manuscript** to address presentation issues. ***[We have uploaded the updated version of our manuscript.]***

---

> ### Author Response · Authors · 2023-11-21
> **Additional Results (new Appendix E)**
>
> ### New Experiments
>
> We conduct experiments on larger models and demonstrate GOT-D's potential in enhancing the zero-shot learning capabilities of LLM. **We evaluate OpenAI's GPT-2 XL (1.5B) and Eleuther AI's GPT-neo (2.7)**, which are widely used in zero-shot learning research. Our analysis encompasses two benchmark tasks: AG News, a text classification challenge focusing on news categorization, and BoolQ, a question-answering dataset involving natural yes/no questions.
>
> For GPT-2 XL whose pre-training data is from a single domain–OpenWebTextCorpus(OWTC), **we use the same data as the candidate dataset**. This setting is the same as the NLG task in Section 3.2. Since all data selection methods (GOT-D, DSIR, curated-TAPT (TAPT with a curated dataset, TAPT/c)) **select from the same candidate dataset**, we are able to fairly compare the performance of each method without the confounding from different pre-processing steps, **providing an ablation study.** GPT-neo is pre-trained on The Pile dataset. We construct a substitute candidate dataset with samples from 7 domains. This setting is the same as NLU tasks in Section 3.3/3.4.
>
> The data selection procedure is similar to the NLG task in Section 3.2. Given a few thousand **unlabeled** training samples (5K for AG News and 9K for BoolQ) as the target data, we test different data selection methods (GOT-D, DSIR, curated-TAPT (TAPT with a curated dataset, TAPT/c)) select samples from the candidate dataset to pre-fine-tune the model. Then, without further fine-tuning, we test the zero-shot classification accuracy of the pre-fine-tuned model on target tasks and measure the performance improvements gained from each data selection method.
>
> **The proposed method establishes a performance gain of 13.9% on AG News and 6.6% on BoolQ after pre-fine-tuning with 40k samples, visibly outperforming baseline methods.**
>
> ---
>
> ### Ablation Studies
>
> Further, with the settings well aligned, **we also ablate on the effect of choices of embedding space for computing OT distance.** We tested embedding samples with **distilled-BERT, sentence-transformer, and BERT-tokens**. Lightweight and fast, the popular sentence-transformer uses a pre-trained all-MiniLM-L6-v2 model with 22M parameters as the backbone. It embeds up to 6 million samples/hours on a single GPU and is sometimes considered a 'default' option for sentence embedding in many NLP tasks. Token space isn't a proper embedding for OT (e.g., the distance on token space is not invariant to paraphrase). We are only listing it here for comparison.
>
> As can be seen from Table 17, the performance of sentence-transformer is mostly on par with distilledBERT. **It suggests the choice of embedding space isn't a critical part of the data selection pipeline and any reasonable embedding space should work.**
>
> ---
>
> ### Visualization of Selected Data
>
> Besides, in the additional experiments with the “AG News” task where all the settings are best aligned, **we conduct visualization of samples selected by each method and the distribution of target task data and pretraining/candidate data**, offering insights on how the proposed method works differently from the distribution-matching approaches. As shown in Figure 16, the comparison shows a clear contrast. **Both DSIR and TAPT/c select samples that match the distribution of the target task data which has a high overlapping with the distribution of the candidate data where the model is already pre-trained on.** Thus, with such a small data budget, the information gain provided from pre-fine-tuning on these samples is naturally marginal. In contrast, **GOT-D selects predominately formal business news which is highly underrepresented in the candidate dataset but important for the target task.** Pre-fine-tuning the model with these samples provides a more direct benefit in aligning the model with the target tasks, effectively validating the idea of this work.

---

> ### Author Response · Authors · 2023-11-21
> **Paper Edits**
>
> We made a range of revisions to the manuscript and uploaded the updated version.
>
> - **We revamped the main methodology section**–Section 2.3, DATA SELECTION FOR FINE-TUNING to improve its rigor and clarity and the general quality of the paper. We formulate our proposed approach into a formal lemma/theorem with proofs and remarks, replacing the previous derivation-with-elaboration.
>
> - A number of detailed revisions to the presentation issues kindly pointed out by the reviewers. We strive to improve the clarity of the manuscript and ease the burden on the readers.
>
> - Additional experiments are provided at the end of the Appendix. We aim to provide it as part of the main paper.

---

### Author Response · Authors · 2023-11-23
**End of discussion period and thanks note ☺️.**

Dear Reviewers and Area Chair,

The authors express their sincere appreciation for your valuable time and efforts spent reviewing this paper and helping us improve it ☺️.

Despite the end of author discussions, we strive to consistently improve the paper. Please don't hesitate to let us know anytime if you have any further comments or feedback.

If there are questions from the discussion between the reviewers and the AC, we would be more than glad to know and could take the last few moments today to respond to them and help explain.

Many thanks,\
Authors of Submission530

---

### Meta-Review · Area_Chair_bZNu · 2023-12-02

**Metareview:**

The paper presents a novel method, GOT-D, for data selection at the pre-fine-tuning stage to enhance the task adaptation of LLMs. GOT-D leverages OT distance as a measure to prioritize samples that shift the pre-training distribution closer to the target fine-tuning distribution. This approach differs from continued pre-training and focuses on using a smaller data selection budget more effectively. The paper's experiments on diverse tasks, including domain adaptation, the GLUE benchmark, and fairness evaluations, demonstrate GOT-D's efficacy.

Some reviewers noted that the improvements over previous methods are marginal, which might question the practical significance of the approach. However, the authors point out that the gain is achieved with a smaller data budget. Further experiments with larger models confirm the approach's scalability and effectiveness.

**Justification For Why Not Higher Score:**

The reviewers point out several concerns about its initial version.

Innovative but Marginal Improvements: While the GOT-D method introduces an innovative approach in data selection for pre-fine-tuning LLMs, the improvements it offers over existing methods are described as marginal by some reviewers. This suggests that while the paper contributes to the field, the extent of its impact might not be groundbreaking enough to warrant a higher presentation category.

Limited Comparison with Baselines: The paper's comparison with a limited number of baseline methods (DSIR and DAPT) may only partially establish its superiority. A broader comparison with various methods might have strengthened the paper's position.

Presentation: The initial lack of detailed information about candidate datasets and OT calculations, although later addressed, might have been addressed.

**Justification For Why Not Lower Score:**

The paper introduces GOT-D, a novel method for data selection in pre-fine-tuning LLMs, which is a contribution to the field. The paper presents experiments across diverse tasks, such as domain adaptation, the GLUE benchmark, and fairness evaluations, showing the efficacy of the GOT-D method. These positive results (although marginal gain in some cases) support the paper's claims and provide a solid foundation for its acceptance. The authors responded constructively to reviewer feedback, including conducting additional experiments with larger models and addressing concerns about the presentation and depth of analysis.

Following extensive discussions with all reviewers, there is a unanimous agreement that the paper makes a notable technical contribution. However, there are concerns regarding the experiments, which are perceived as flawed. It is recommended that additional experiments and ablation studies be included to strengthen the paper. Reviewers want to see more improvements for proposed results.

This paper currently sits at the borderline — its approach is commendable, but its experimental outcomes are less solid. From the AC's perspective, I propose placing greater emphasis on the approach over the experiment performance. This is based on the rationale that fine-tuning hyperparameters, with adequate computational resources, can potentially yield improved results. Hence, I ultimately advocate for the acceptance of the paper.

---

### Decision · Program_Chairs · 2024-01-16

Accept (poster)